# Accelerated and Natural Aging of Cellulose-Based Paper: Py-GC/MS Method

**DOI:** 10.3390/molecules27092855

**Published:** 2022-04-30

**Authors:** Alexander Kaszonyi, Lívia Izsák, Milan Králik, Michal Jablonsky

**Affiliations:** 1Department of Organic Technology, Catalysis and Petroleum Chemistry, Faculty of Chemical and Food Technology, Slovak University of Technology in Bratislava, Bratislava I, 81237 Bratislava, Slovakia; livia.izsak@stuba.sk (L.I.); milan.kralik@stuba.sk (M.K.); 2Department of Wood, Pulp and Paper at the Institute of Natural and Synthetic Polymers, Faculty of Chemical and Food Technology, Slovak University of Technology in Bratislava, 81107 Bratislava, Slovakia; michal.jablonsky@stuba.sk

**Keywords:** paper aging, Py-GC/MS, pyrogram, levoglucosan, levoglucosenone, furfural, acetic acid

## Abstract

Samples of papers artificially (2 to 60 days) and naturally (10, 45, and 56 years) aged were studied by the Py-GC/MS method to identify decomposition products. Possible reaction scenarios for cellulose degradation were developed. One of the degradation products is acetic acid, which can (auto)catalyze the cleavage of cellulose β(1→4)-glycosidic bonds of cellulose polymer chains. However, during 20 s of Py-GC/MS analysis, temperatures of up to 300 °C did not significantly increase or modify the formation of decomposition products of paper components. At 300 °C, the amount of several cellulose decomposition products increased regularly depending on the number of days of artificial aging and natural aging, demonstrated mainly by the generation of 2-furancarboxaldehyde, 5-hydroxymethylfurfural, and levoglucosan and its consecutive dehydration products. No correlation between the amount of lignin decomposition products and the time of aging was found when the pyrolysis was performed at 300 °C and 500 °C. Compounds present in the products of decomposition at 500 °C bear the imprint of the chemical composition of the sampled paper. Pyrograms taken at 300 °C using the Py-GC/MS method can give additional information on the changes in the chemical structure of paper during natural or artificial aging, mainly about the cleavage of β(1→4)-glycosidic bonds during aging.

## 1. Introduction

The degradation of cellulose-based paper is a complex multistimulus process. The raw material used for paper production (type of wood fibers, nonwood fibers, and other components (fillers, additives, pigments)), pulp making (kraft pulping, sulfite, soda, organosolv pulping), sizing (especially aluminum sulfate–rosin mixtures used in the past), initial acidity, and content of lignin and metal ions have the most important influence on paper degradation. During storage of lignocellulosic goods, e.g., books, the degradation process is stimulated by factors such as moisture and acidity; increased temperature; intensity of light; mechanical stress, including folding history and abrasion; and biological contamination [1,2]. Storing paper in a dry and dark room, without mechanical stress, is the simplest way to decrease the rate of paper degradation. However, chemical degradation will continue depending on the acidity, moisture content, temperature, presence of redox catalysts, and oxygen concentration [1,2]. During storage, the acidity of paper increases due to the increasing concentration of volatile organic acids, such as acetic acid and formic acid, formed by chemical degradation of the material, especially hydrolysis and oxidation of polysaccharides in the paper, particularly cellulose polymer chains. The process can be considered autocatalytic [3,4]. The detailed mechanism of acetic acid formation can be found in the recent paper from Potthast et al. [5]. Other volatile degradation products such as formaldehyde, furfural, and vanillin represent decomposition products of cellulose and lignin [4,6,7]. However, in old books in libraries, many other volatile compounds were identified (aromatics, 2-ethyl-1-hexanol, long-chain aldehydes, alkanes, and esters of fatty acids) [8]. The wide spectrum of decomposition products can be ascribed to the presence of aluminum and sulfate anions incorporated in the paper in the process of sizing using the alum–rosin procedure [9]. Although oxidation is not such an acute problem as acid hydrolysis due to alumina, it is usually accelerated by the presence of transition metals in the paper [10,11]. Other factors influencing the decomposition are solvents used in printing inks and pigments, very often containing iron species.

The Py-GC/MS method is suitable for the analysis of volatile compounds present in the paper and formed during paper aging or by thermal decomposition of paper. Characterization of degradation products, design, and description of degradation mechanisms of individual paper components can be studied using an analytical pyrolyzer coupled with a gas chromatography–mass spectrometry set-up [12,13,14,15]. The main pyrolysis products of cellulose, cellobiose, and glucose include levoglucosan and levoglucosenone; furans, e.g., 5-hydroxymethyl furfural and furfural; and small molecular substances such as CO_2_, 1-hydroxy-2-propanone, acetaldehyde, acetic acid, and propanoic acid [15]. The Py-GC/MS equipment allows treating a sample of organic matter by dropping it into a pyrolytic microreactor preheated to temperatures of up to 1000 °C. Volatilized compounds are then transported by the carrier gas to an efficient chromatographic column, where components are separated and identified based on their mass spectra. By applying this technique, chemical structures of decomposition products can be identified with the aim of characterizing the chemical features of the paper. This information also enables the determination of different organic materials such as sizing agents, ink components, waxes, resins, oils, binders, pigments, and dyes in paper [16,17,18]. The chemical composition of the original components of a sample can be reconstructed by detailed interpretation of the molecular profile of thermal degradation products. However, due to a huge number of consecutive and parallel reactions, the interpretation of complex matrices is challenging.

Basic components of lignocellulosic materials start to decompose at different temperatures during heating in inert gas: amorphous hemicellulose at around 250 °C, crystalline cellulose fibers at around 300 °C, and lignin at 350 °C [19]. All the mentioned materials are obviously present in the studied newsprint papers, and all of them can be chemically changed/degraded during shelf time. It is worth mentioning that a micropyrolytic approach was also successfully applied for the characterization of products from fast micropyrolysis of lignocellulosic materials [20], where the main focus was on the characterization of bio-oil obtained by pyrolysis.

The aim of this work was to study the processes of chemical degradation of paper during aging by applying the Py-GC/MS method for the determination of formed volatile organic compounds.

## 2. Results

The composition of volatile organic compounds formed during natural and artificial aging of a paper was studied by Py-GC/MS methods. Paper sheets were artificially aged for 0, 2, 15, 20, and 60 days, and some of them were naturally aged in laboratory conditions for 10 years without mechanical stress and light in a closed aluminum foil envelope. Paper samples of 1 mg were dropped into the pyrolyzer preheated to 200, 250, 300, or 500 °C, and evaporated compounds were collected by the splitless method at the beginning of the capillary column for 20 s. After flashing out the injection port and pyrolyzer by opening the split valve, the collected volatile compounds were analyzed by the GC/MS method. When comparing data obtained at 200, 250, and 300 °C (Figure 1 and Figure 2), a reasonable evolution of volatile and decomposition products was found at 300 °C. Therefore, we took this temperature as a starting point for elucidation of sample degradation analysis.

In the first series of experiments, volatile compounds released from practically new (unused) newspaper were identified. As it is seen in Figure 1, the first traces of volatile compounds of unknown origin were recognized at 300 °C. At 500 °C, the sample was decomposed to the expected decomposition products of cellulose and lignin. This result is in agreement with literature data [19] concerning the stability of the main lignocellulose components.

Acetic acid is a typical product of paper aging [1] and can catalyze the decomposition of paper components. In another experiment, the ability of acetic acid to allow the formation of a detectable amount of volatile compounds within 20 s of sampling time in the Py-GC/MS was verified. A new newspaper sample and old textbook paper (manufactured in 1976) were saturated with acetic acid in its vapors in a closed vessel under laboratory conditions. In the measured pyrograms, no new components were found, except for acetic acid, and the differences in the amount of evaporated compounds are within experimental error (Figure 3); i.e., acetic acid present in the sample during Py-GC/MS analysis up to 300 °C did not increase or modify the formation of decomposition products of paper components significantly.

No immediate effect of acetic acid on decomposition products indicates a more complicated autocatalytic reaction pathway of cellulose chain decomposition. 

In comparison to new newspaper, a significantly higher amount of volatile compounds was determined in the case of papers after 10-year or longer natural aging (Figure 2 and Figure 3) at 250 °C and 300 °C. The identified compounds, which were practically the same at both temperatures, are listed in Table 1. However, evaporated amounts of compounds at 250 °C are significantly lower. Virtually all compounds determined are formed by the decomposition of cellulose and/or lignin, except for nonanoic acid, palmitic acid, and dibutyl-phthalate, which were probably added during paper manufacturing. In the case of paper from old textbooks, a significantly lower amount of lignin components was found, which indicates a lower content of lignin in the paper used for textbooks.

When the paper sheets at the beginning of natural aging were artificially aged, the amount of some cellulose decomposition products regularly increased with the number of days of artificial aging. At the same time, the amount of lignin degradation products varied randomly without a regular increase (Figure 4a–f).

Before discussing the effect of natural aging on the products of cellulose and lignin decomposition, it should be noted that not all decomposition products are able to elute from the column at the experimental conditions and thus are not detected by the MS detector. Among the lignin degradation products, only compounds containing one aromatic ring were detected. These compounds contain zero, one or two –OH groups; one or two CH_3_O- groups; and zero or one alkyl group with maximally three carbons. The detected decomposition products of cellulose contain from one to six carbons of one glucose unit of cellulose. No products containing carbons of two or more glucose units were detected. Similarly, the volatility of compounds with four or more –OH groups is too low to allow elution from the used column.

When the samples were pyrolyzed at 500 °C, no correlation was found considering the decomposition product of cellulose or lignin and the time of artificial aging. The area of levoglucosan (LG) and other cellulose decomposition products was approximately the same in the range of experimental error for all times of artificial aging. 

However, when the samples were pyrolyzed at 300 °C, the highest increase in the area of decomposition products was detected for levoglucosan (LG), which is formed mainly by hydrolysis or substitution of the last β-glycosidic bonds between glucose units at the end of cellulose macromolecules. Artificial aging for 2 and 5 days increased the area of LG nearly 5-fold, while 20 and 60 days increased the area by almost 8-fold. In papers naturally aged for 45 and 56 years (papers from 1976 and 1965, respectively), the area of LG increased nearly 20-fold compared to those naturally aged for 10 years. The amount of levoglucosenones, formed by dehydration of levoglucosan, was increased by artificial aging up to 2.5 times. The amount of FF and HMF, formed by the dehydration of the glucose unit of cellulose, increased up to 12- and 4-fold, respectively. 

In several experiments, it was determined whether the decomposition products were formed in the microreactor of the pyrolyzer (during 20 s of evaporated product collection from samples at 300 °C) or had already formed during artificial or natural aging; i.e., compounds determined during heating in the pyrolyzer were simply evaporated from samples without cellulose decomposition. As practically all decomposition products can be washed out by water and methanol, samples of papers from old textbooks were carefully washed with water and then with methanol and dried at room temperature. Pyrograms before and after washing are compared in Figure 5. As can be seen, washing removed a very small to negligible portion of the decomposition product. The amount of furfural, which was also recognized by other authors studying paper aging (e.g., [1]), is significantly lower. Strilc et al. [4] confirmed that furaldehyde is a valuable source of information and there is a correlation between furaldehyde emissions and the pH of the cellulosic environment. From our measurement follows that the detected decomposition products are formed mainly during the thermal decomposition of paper at 300 °C. This temperature is at the beginning of cellulose decomposition determined by DTG experiments [19]. A new paper sheet, typically with long macromolecules of cellulose polymer chains stabilized in cellulose bundles by hydrogen bonds (consequently with a low amount of reducing or non-reducing ends of the molecules) is able to withstand this temperature during the Py-GC/MS analysis. The amount of formed volatile decomposition products during 20 s of pyrolysis is negligible. During natural or artificial aging, macromolecules are split/shortened; thus, significantly more molecular ends are formed and can undergo further decomposition. The original tight network of the hydrogen bonds is reorganized, and glucose units are kept by fewer hydrogen bonds in the fiber. The significantly increased amount of macromolecular ends and decreased amount of hydrogen bonds result in a markedly higher rate of decomposition product formation.

As it is known, the most significant changes during paper aging occur in the structure of cellulose bonds, consequently affecting the properties of microfibrils, macrofibrils, cellulose fibers, and overall mechanical strength. Changes in color due to the creation of chromophores and the generation of colored chemical compounds were also observed in the previous research [21]. Decomposition of cellulose bonds and the following reactions are catalyzed by acids, which can be residual or newly formed during aging. Of the newly formed acids, the most important is acetic acid [1].

Based on our results and published data, the following reaction network of cellulose decomposition and the formation of cellulose decomposition compounds is proposed:

Protons, which are “the real catalyst” in acid-catalyzed reactions, can migrate very efficiently both along the hydrogen bond network and from one cellulosic chain to another (Figure 1).

Then, protons on the oxygen of β(1→4)-glycosidic bonds (further also etheric O) can decrease the density of electrons on both carbons connected with this etheric oxygen atom (O) and catalyze hydrolysis or substitution of etheric O at ambient temperature. The content of water in common paper is about 3–10 wt.%, which is theoretically enough to hydrolyze all β(1→4)-glycosidic bonds in cellulose. Hydrolysis of any β(1→4)-glycosidic bond provides one reducing and one non-reducing end of the cellulose macromolecule (Figure 2). In the inner macromolecules of the polymer cellulose chain, where the presence of water is less probable, protons on etheric O can catalyze the substitution of this oxygen by oxygen of the hydroxymethyl group (C6 in glucose). The formation of the levoglucosanic end of the macromolecule can be observed (Figure 3).

These reactions are supported mainly by mechanical stress at lower temperatures, and at 300 °C, they are also supported by the thermal motion of atoms in the molecules. The substitution reaction can be catalyzed by protons not only on the surface of cellulose fibers, but also in internal macromolecules, where water molecules are not present, but protons can enter through a network of hydrogen bonds (Figure 1 and Figure 3). 

Further hydrolysis or substitution on the levoglucosanic end produces levoglucosan, similarly to the substitution on the non-reducing end of the macromolecule (Figure 3). Hydrolysis or substitution on the reducing end produces glucose, similarly to the hydrolysis on the non-reducing end of the macromolecule. Proton-catalyzed cleavage of β(1→4)-glycosidic bonds can take months and years of natural aging at ambient temperatures or days of artificial aging at nearly 100 °C. When a higher temperature is applied (e.g., 300 °C as in our Py-GC/MS experiments), these reactions proceed at a second timescale. Levoglucosan and its derivatives formed by dehydration or isomerization are the main decomposition products of cellulose, as determined by the Py-GC/MS method applied in this work.

A proton on the acetalic oxygen in the glucopyranose ring can catalyze the opening of this ring and the formation of a reactive aldehydic end of the molecule (Figure 4) or glucose, which can undergo further retro-aldol condensation reactions (Figure 4 and Figure 5). 

Products of deeper cellulose decomposition, which can be formed from glucose units or from LG by isomerization and dehydration reactions, are summarized in Figure 5 and Figure 6. All final compounds from Figure 5 and Figure 6 were found in the pyrograms of decomposition products. In Figure 5a, possible mechanisms of HMF and FF formation are presented. Figure 5b shows the proposed retro-aldol condensation and the formation of hydroxyacetaldehyde and consecutively of acetic acid.

Levoglucosans (LG in Figure 6) have three –OH groups with several possible spatial orientations, which results in low volatility (reflected in long elution time and a wide peak in the pyrogram). By decreasing the number of –OH groups and molecular weight, elution times of their dehydration products (progressively: levoglucosene, levoglucosanone, and levoglucosenone) are shortened (Table 1). All the cellulose decomposition products with more –OH groups and several possible spatial orientations are eluted in the wider peak.

Some decomposition products of cellulose and lignin, identified from mass spectrum libraries, are summarized in Figure 7.

## 3. Materials and Methods

### 3.1. Raw Material

Commercial wood-containing newsprint paper (grammage 45 g.m^−2^, pH determined by cold extraction: 4.5–5.0), containing mechanically bleached groundwood (55%), bleached sulfite pulp (20%), recycled fibers (15%), and clay (10%), was used in all accelerated and natural aging experiments. 

In addition, commercial newsprint paper (grammage 80 g.m^−2^, Mondi SCP a.s., Ružomberok, Slovak Republic) was analyzed on the third day after manufacturing for comparison. 

In addition, two samples of paper from textbooks printed in 1965 and 1976, which were naturally aged in the office library, were analyzed.

### 3.2. Accelerated Aging at 98 °C

Samples of paper were conditioned according to TAPPI T402 om-93 at 23 ± 1 °C and 50 ± 2% relative air humidity (RH) [1,2]. The samples were subsequently aged according to ISO 5630 (2005), using the modified temperature of 98 °C ± 2 °C (instead of 100 °C) and 50% RH, corresponding to paper humidity of 4–5%. Twenty sheets of paper (A4 format) were encapsulated inside a PET/Al/PE composite foil (Tenofan Al/116S) by sealing off all four edges using a Polystar 30D impulse tong sealer (Rische & Herfurth, Hamburg, Germany). This bag was put into another PET/Al/PE bag and completely sealed off. Batches of samples were put into a thermostat for 0, 2, 15, 20, and 60 days and kept at the temperature of 98 ± 2 °C according to ASTM D 6819 (2002): Standard test method for accelerated aging of printing and writing paper by dry oven exposure apparatus, in which sealed glass tubes were replaced by a composite foil made of polyethylene/aluminum/polypropylene (Tenofan Al/116S). Humidity inside the bag during accelerated aging was 50 ± 2% (at 20 °C) and free air volume in the bag was 5 ± 1 mL. After accelerated aging, some of the papers were left in the PET/Al/PE bag for the next 10 years under laboratory conditions before analytical pyrolysis by the Py-GC/MS method.

Some important properties of artificially aged paper are summarized in Table 2. 

### 3.3. Pyrolysis Process, Effect of Temperature

In order to study the influence of the pyrolysis temperature on the VOC content and fragmentation of paper components, samples of approximately 1 mg were taken and dropped into the pyrolyzer preheated to 200, 250, 300, or 500 °C, and the vaporized compounds were analyzed by GC/MS. Representativeness of the entire sample and sensitivity of the analysis were examined using the chromatogram–pyrogram of several samples.

### 3.4. Py-GC/MS Analysis

Analysis of the microsamples was carried out in a Py-GC/MS equipment, which allows evaporation and eventual thermal decomposition of organic matter from the samples at temperatures of up to 1000 °C. Volatile/decomposed substances were transported to an efficient chromatographic column, separated, and identified based on their mass spectra. The pyrolytic microreactor (Frontier Multi-Shot Pyrolyzer, model EGA/PY-3030D) was directly connected to the injector of a gas chromatograph coupled to a mass spectrometer (Shimadzu GC-MS-QP2010 Ultra). For chromatographic separation, an Ultra ALLOY-5 capillary column (30 m × 0.25 mm × 0.25 μm) was used. The carrier gas was helium (99.995%) with a constant flow of 1.0 mL min^−1^ at the pressure of 50 kPa. The injector and transfer line temperatures were set to 300 °C. The injector worked in splitless mode for 20 s and then in split mode (split ratio 1:10); a solvent cut of 1 min was selected. The oven temperature program started at 50 °C, keeping this temperature for 5 min and then increasing it by 10 °C.min^−1^ to 280 °C, which was held for 30 min. Operation conditions for electron impact (EI) mass spectrometer were as follows: ionizing voltage of 70 eV and SCAN range from m/z 27 to m/z 550. Structural identification of the fragments was performed by comparison with the NIST 11 and Wiley MS library and literature data from previous studies. The GC/MS technique could not provide a direct quantitative analysis of the compounds due to the complexity of pyrolytic products and the lack of commercially available standards. However, the chromatographic peak area of a compound is considered linear with its quantity, and the peak area percentage provides information about its relative content in the detected products. Pyrograms presented in the figures are dependencies of MS TIC (total ion current) vs. elution time in minutes.

The transport of volatiles from the microreactor to GC columns after 20 s of splitless mode was not stopped. Less volatile compounds were gathered at the beginning of the GC column for 5 min at 50 °C. In the preliminary experiments, we found that a long splitless time results in strongly overlapping and tailing peaks with potential clogging of the needle at the bottom of the pyrolytic quartz tube. Here, 1 mg samples and 20 s splitless time have shown to be a reasonable optimum.

To clean the GC column, the first activity before a routine analysis was a blank analysis. Peak heights in the blank analysis were lower than 1% of normal peak heights. The blank analysis after normal analysis was without peaks (see, e.g., in Figure 1, experiment at 200 °C).

An error for determination of components equal to ±4.5 % was estimated from four repeated analyses at 300 °C of paper after 10-year natural aging, taking into account the chromatographic area for furfural. The reproducibility is also well seen in Figure 2 and Figure 5.

## 4. Conclusions

As a result of artificial aging, there have been significant changes in the properties of the paper (Table 2). VOC products for these samples were analyzed, and possible reaction scenarios for cellulose degradation were developed. In particular, the results of the analysis after artificial aging are very important because they provide important information about VOC production and simulate what VOC products are also formed for very old (naturally) aged papers after pyrolysis. 

Pyrograms taken at 300 °C by the developed Py-GC/MS method provide additional information on the changes in the chemical structure of paper during natural or artificial aging. Decomposition products in the pyrogram are formed by retro-condensation and dehydration reactions. Acetic acid can catalyze the cleavage of cellulose β(1→4)-glycosidic bonds during natural and artificial aging because of the available log time. However, during 20 s of the Py-GC/MS analysis up to 300 °C, acetic acid did not increase or modify the formation of decomposition products of paper components significantly. This implies a much more complicated reaction pathway connected with the movement of protons on the cellulose chains, which is outlined by the above reaction schemes. 

When the samples were pyrolyzed at 500 °C, no correlation was found between the decomposition products of cellulose or lignin and the time of artificial aging. However, the compounds present in the decomposition product bear the imprint of the chemical composition of the sampled paper. The compounds formed from cellulose, lignin, and additives were identified. At 300 °C, the amount of several cellulose decomposition products regularly increased with the number of days of artificial aging and/or natural aging, which is evident mainly for FF, HMF, and levoglucosan and its consecutive dehydration products. Under the same conditions, no decomposition products were detected in the case of new paper (on the second day after manufacturing). 

## Data Availability

Data presented in this study are available on request from the corresponding author. The data are not publicly available as the research is still in progress.

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
