# Peer review of "Accelerated and Natural Aging of Cellulose-Based Paper: Py-GC/MS Method"

_molecules, 2022, doi:10.3390/molecules27092855_

Round 1

Reviewer 1 Report

The manuscript describes the use of Py-GC-MS to characterise samples of papers artificially and naturally aged. The subject of the manuscript is interesting and within the scope of Molecules.

I have spotted some flaws that prevent the manuscript could be published in its present form:

  1. it is not clear on which bases a temperature of 300°C was selected for the experiments. Preliminary TGA analyses would have been useful to obtain information regarding the thermal stability of the samples. Please, add information regarding this point.
  2. At 300°C paper undergoes thermal desorption more than pyrolysis. This means that you can monitor only the molecules that are free and not involved and bonded under a polymeric form, such as cellulose. Please, correct the manuscript on the bases of this information.
  3. I have some concerns regarding how the experiment have been carried out: is the time of desorption sufficient for a complete and representative desorption of the free analytes from the paper samples? Several of the compounds formed during the degradation of paper are polar and have a low volatility. I not sure that 20s are enough for a quantitative recovery of all the free analytes. How have you selected such time?
  4. The polarity of your analytes can induce memory effects that can alter the evaluation of the data. Have you look for such effects in your experiments? Are blank samples routinely run?
  5. Are your analyses reproducible? Which is the CV%?
  6. Compounds identified in the chromatograms have to be list in a table reporting not only the retention time but also an identification number (1, 2, 3….n). The identification number have also to be used to indicate the various compounds in the chromatograms.
  7. The variation of the various compounds at the increasing of the aging time is not evident. Could you treat the experimental data (chromatographic areas) so that variation in the abundance of certain compounds could be evident?
  8. It is not clear which articles have been used to identify the compounds by their mass spectra. Please, add the relevant literature.

Author Response

Response to reviewer 1 is in the Word file.

Reviewer 2 Report

Dear Authors and Editors,

the persent work (if accepted in the end) should be completely reworked and extended to meet the criteria of Molecules (Q1) journal.

The work lacks many aspects of a good article unfortunatelly although it applies state-of-the-art methodology and the paper in itself seemed to be interesting, but at the end it came up with too general conclusions and results.

In particular: The abstract lacks a clear definition of a rationale, and research objectives. One gets no clue why the authors actaully did the research. The one sentece at the end of chapter 1 is way too general. Such research have already done, I think.

Interpretation of tables and figures is a nightmare. Table1 lists the identified compounds but there is no annotation of the peaks in figures (e.g. in 2, 3, etc.) where these compounds can be found. For example: please seach for the peak of  vanillin (tr: 17.79 min) in figure 2. Good luck! This is very confusing and does not help the reader for interpreation and must be improved.The same applies for figures 4, you do not know actually what you should be looking at precisely.

Scheme1 is a textbook figure not for an article I think.

Other reaction schemes are good and well detailed and easily followable.

Materials and methods are well described, yet some details lack or need to be addressed: 

L374: what does it mean "several samples". This whole sentence does not make any sense this way please rephrase!

L 392-398: For regular liquid injection/splitting the peak areas can be calibrated for concentration in GC(MS) yet, for headspace and pyrolysis techniques you sample an equilibrium headspace thus the peaks you get are not eveidently "linear with its quantity" as you suppose but are influenced by many factors (conditions, temperature, adsoprtion, desorption equlibiria). PLease cite here a reference that that proves that actually your statement is true ALSO for py-Gc-MS.

Because the aims and objectives were way too general so are the conclusions. There even sentences that make completely no sense: 

L412-413, L418-420.

One really gets the impression that there was really no real objective of the research, only measureing some samples and evaluation the results and speculating over the reaction schemes. If this was the aims, then let it be, but then it whould be stated that the real focus was the implementation of reaction mechanisms. Yet again, in this case the novely of the results have to be justified 

Author Response

Response to reviewer 2 is in the Word file.

Reviewer 3 Report

Dear authors,

I have reviewed your paper, but I have found some unclear facts:

line 29>> Kraft pulping uses NaOH and Na2S.. it is true that the withe liquor used as pulping agent does contain some sodium sulphate but the question is how much of this anion could be found in bleached pulps used for papermaking?

line 38>> Please specify references in a much more clear format.

line 41 >> How does the mechanism of cellulose to acetic acid and/or formic acid does look like?

line 48>> How about sizing with AKD? -Alkyl ketene dimer? it's been used since 1950's! https://doi.org/10.1016/j.arabjc.2012.04.019

line 106 (and further more) >>> pyrograms graphs must be imporved ...they look as copy paste from a proprietary software.. such as that of an equipment or so...

lines 337>>> you have used two types of papers in your study.. but you have not determined the chemical composition. How do you know that they are similar or alike? on the other hand, both types of paper are considered to be mostly composed of mechanical pulp fibers (over 55%) or is just my supposition?  Have you thought including in your study the paper obtained from 100% bleached Kraft pulp to show up the differences and the similarities in degradation pathway? This could be used as control experiment.

line 363 >> table 2 describes the properties of some aged papers. Most likely they have a different  chemical composition than yours.

Author Response

Response to reviewer 3 is in the Word file.

Round 2

Reviewer 1 Report

The manuscript has not been improved and several issues have not been addressed by the authors. 

Author Response

Response 1:  The manuscript has been improved and all issues have been addressed by the authors in the previous responses to reviewers (Responses 1). Please check the last version of the manuscript text.

Reviewer 2 Report

Dear Authors and reviewers,

the authors have significantly improved their work and with the extensions and explanations it can be accepted for publication. Also the rebuttals are clearly explained.

Author Response

Many thanks for the final recommendation.

Reviewer 3 Report

Dear authors,

I have analyzedd your paper. It still has some issue. One of them is concerning the sulfate anion. I hope this time I have a clearer question>>>what is the influence of sulfate anions originating from Kraft process (only) on paper aging? I think that the link between the pulping process and further bleaching and the aging behavior is different and incompletely presented. Please explain.

The other issue concerns  the method used for establishing the molecular weight of cellulose from your paper samples. Even if cited work is used in the table, a brief description could be of use.

All graphs could be improved by removing the grid lines.

Kind regards.

Author Response

Response to Reviewer 3 Comments is in the attached file.
